# The unspoken reality of gender bias in surgery: A qualitative systematic review

Wen Hui Lim[1], Chloe Wong[1], Sneha Rajiv Jain[1], Cheng Han Ng[1], Chia Hui Tai[2], M. Kamala Devi[3], Dujeepa D. Samarasekera[4], Shridhar Ganpathi Iyer[5,6], Choon Seng Chong[1,2]*

1 Yong Loo Lin School of Medicine, National University of Singapore, Singapore, Singapore, 2 Division of Colorectal Surgery, Department of Surgery, University Surgical Cluster, National University Hospital, Singapore, Singapore, 3 Alice Lee Centre for Nursing Studies, Yong Loo Lin School of Medicine, National University of Singapore, Singapore, Singapore, 4 Centre for Medical Education, Yong Loo Lin School of Medicine, National University of Singapore, Singapore, Singapore, 5 Division of Hepatobiliary & Pancreatic Surgery, Department of Surgery, University Surgical Cluster, National University Hospital, Singapore, Singapore, 6 Liver Transplantation, National University Centre for Organ Transplantation, National University Hospital, Singapore, Singapore

* choon_seng_chong@nuhs.edu.sg

**Data Availability Statement:** All relevant data are within the manuscript and its Supporting Information files.

**Funding:** The author(s) received no specific funding for this work.

## Abstract

### Objective

This study was conducted to better understand the pervasive gender barriers obstructing the progression of women in surgery by synthesising the perspectives of both female surgical trainees and surgeons.

### Methods

Five electronic databases, including Medline, Embase, PsycINFO, CINAHL and Web of Science Core Collection, were searched for relevant articles. Following a full-text review by three authors, qualitative data was synthesized thematically according to the Thomas and Harden methodology and quality assessment was conducted by two authors reaching a consensus.

### Results

Fourteen articles were included, with unfavorable work environments, male-dominated culture and societal pressures being major themes. Females in surgery lacked support, faced harassment, and had unequal opportunities, which were often exacerbated by sex-blindness by their male counterparts. Mothers were especially affected, struggling to achieve a work-life balance while facing strong criticism. However, with increasing recognition of the unique professional traits of female surgeons, there is progress towards gender quality which requires continued and sustained efforts.

### Conclusion

This systematic review sheds light on the numerous gender barriers that continue to stand in the way of female surgeons despite progress towards gender equality over the years. As

**Competing interests:** The authors have declared that no competing interests exist.

the global agenda towards equality progresses, this review serves as a call-to-action to increase collective effort towards gender inclusivity which will significantly improve future health outcomes.

## Introduction

Medicine has traditionally been a male-dominated profession and its longstanding asymmetrical gender order has resulted in deeply entrenched structural barriers that hinder a female's advancement [1]. Although huge progress has been made with more female medical graduates [2], increased female representation, and awareness of gender bias in surgery [3,4], up to 66.7% of females still experience discrimination in the surgical workplace [5–7]. Implicit gender bias also remains persistent within the surgical field [8].

On top of bias in selection of surgical residency candidates [9], gender discrimination has deterred females from pursuing a surgical career [10,11]. This has led to an underrepresentation of women in surgery, compromising quality mentorship and shaping a hostile environment which further cements barriers to entry [10]. Gender bias has also contributed to the higher attrition rates, of approximately 25%, in female surgical residents [12]. With fewer surgeons, the workload of remaining surgeons increases, contributing to surgeon burnout [13], and increased medical errors [14]. Furthermore, high dropout rates of female surgeons are concerning, considering that female surgeons offer valuable attributes, including improved surgical outcomes due to better physician-patient communication and provision of more patient-centric care [15,16]. In addition, diverse representation can better meet the needs of a diverse patient population, as some female patients actively choose female surgeons [17].

Despite drastic implications for population health, the surgical sphere continues to reflect gender disparities that stunt the progression of female surgeons. Analysis of qualitative evidence allows the synthesis of different perspectives to yield deeper insights into gender bias in surgery [18]. Hence, we sought to review current qualitative literature on gender discrimination in the surgical workforce to define ways forward in levelling the playing field for females.

## Materials and methods

### Search strategy

This systematic review was conducted in accordance to the Preferred Reporting Items for Systematic Reviews and Meta-Analyses (PRISMA) statement [19]. Five electronic databases, including Medline, Embase, PsycINFO, CINAHL and Web of Science Core Collection, were searched from inception till 9 May 2020. The search strategy is attached in S1: Supplementary File. In addition, references were hand-searched for additional studies. Articles deemed potentially relevant underwent a title and abstract sieve, followed by a full text review for inclusion by three independent authors. The final inclusion of the articles was based on consensus amongst the three authors. (See also below)

### Study selection and eligibility criteria

Authors individually identified studies that met the following inclusion criteria: 1) qualitative or mixed methods methodology, 2) perspectives of gender discrimination from both female and male surgeons or surgical trainees; and 3) studies related to gender bias in the surgical field. Only original, peer-reviewed articles written in or translated into the English language

were considered. Commentaries, letters to the editor, reviews, conference abstracts, and grey literature were excluded. Additionally, only the clinical practice of surgery was considered in this review, excluding those focusing solely on academic surgery or medicine. Three authors independently conducted full text review, and discrepancies on the inclusion were discussed until consensus was reached.

### Data extraction and analysis

Data was extracted and sorted by two authors using a structured proforma. The structured proforma included origin and year of publication, objective, methodology, demographics (occupation, sample size, gender, age) of participants and primary findings from the included articles. Thematic synthesis was employed to review the data, using the Thomas and Harden framework which comprises three stages of detailed synthesis: line-by-line coding of the primary text, construction of descriptive themes, and the development of analytical themes [20]. Repeated reading of primary data was conducted by two authors to identify recurrent ideas to form descriptive themes that were compiled, debated, and categorized until a consensus was reached. Analytical themes were distilled by forming a relational quality among descriptive themes to synthesise perspectives beyond primary data. Discussions were held among authors for clarification and comparison of primary findings and final synthesis.

### Quality assessment

Quality appraisal of included studies was conducted using the Critical Appraisal Skills Programme (CASP) Qualitative Research Checklist and the Standards for Reporting Qualitative Research (SRQR) [21,22]. The CASP Checklist consists of 10 items developed to assess the trustworthiness, relevance and results of published papers. The SRQR consists of 21 recommended reporting standards. Both appraisal tools facilitate transparency in all aspects of qualitative research by formulating a set of guidelines to optimise reporting. Quality assessment was independently conducted by two authors, with disagreements being resolved by discussion with a third author until consensus were reached. The results of quality assessment did not result in exclusion of any studies, but increased the collective rigor of the synthesis [23].

### Results

Electronic search results identified a total of 3,716 articles, 2,948 remained after duplicate removal, and 188 articles were selected for full text review, of which 14 research papers met the inclusion criteria. This is presented in Fig 1. In total, there were 528 participants. There were 300 female participants and 228 male participants. Out of 300 female participants, 151 were surgeons and 149 were residents. Out of 228 male participants, 120 were surgeons while 108 were residents. The age of participants ranged from 32 to 63 years old. The included studies were conducted in eight different countries: six in the United States [24–29], four in the United Kingdom [30–33], two in Australia and New Zealand [34,35], one in Canada [36], and one in Rwanda [37]. The characteristics of the included papers are presented in Table 1.

The quality of included articles by CASP and SRQR can be found in S2 and S3: Supplementary File respectively.

In the thematic synthesis of codes regarding barriers posed by gender bias in surgery, three analytical themes were generated: unfavorable work environment, male-dominated culture, and societal pressures. An overview of the themed analysis of gender bias in surgery is presented in Fig 2. One positive analytical theme on progress towards gender equality was also generated.

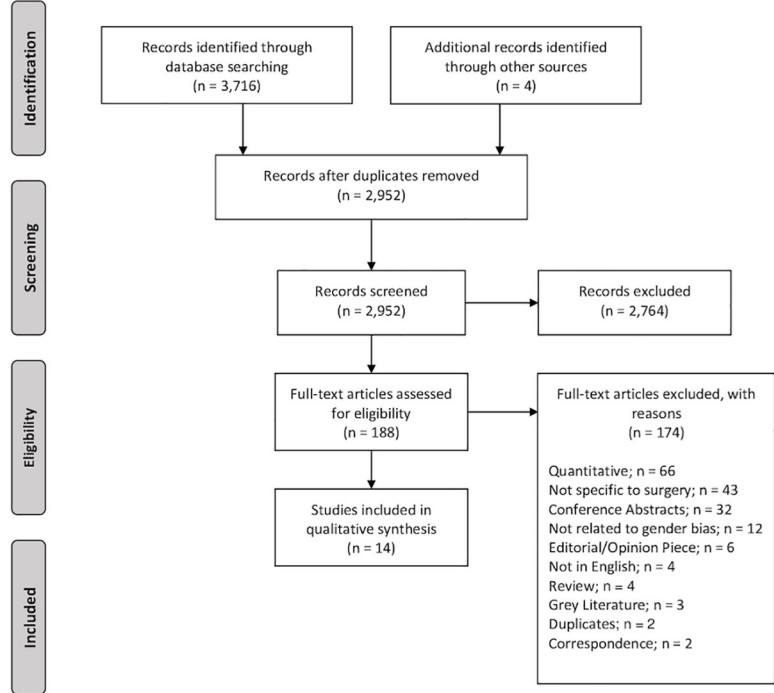

**Fig 1. PRISMA flow diagram.**

## Unfavorable work environment

**Harassment.**   Female surgical residents were exclusively subjected to unwanted and unsettling incidents of sexual harassment, usually by either male colleagues, seniors, or patients [24–26,34,35]. Reported behaviors include inappropriate physical contact, derogatory depictions, suggestive winking and smiling at female trainees, coupled with elbow nudging [24,34,35]. Female surgeons and surgical trainees were also targets of sexualized comments centred on their dressing, physical appearance or demeaning sexual offers [24,27,31,34,35]. Such inappropriate behaviors asserted superiority over female surgeons [26], undermining their professional standings [24,26]. Female surgical residents often reported feeling deeply uncomfortable when confronted with such sexual advances, yet faced uncertainty about reporting due to perceptions that they were being over-sensitive or fear of judgment [24,26,27]. Uncertainty about reporting such acts arose from being told that they were being over-sensitive and thus, female residents worried that their response would be deemed disproportionate [24,27].

**Insufficient support.**   Female surgeons and trainees reported a lack of female mentorship and role models [25,28,34], as there were insufficient opportunities where women could share problems and seek advice [28,34], which made the training process more isolating and discouraging [25,28,34].

When it came to motherhood, female surgeons reported a lack of maternity support, even from female seniors. Senior colleagues explicitly expressed that they would not support surgical residents who chose pregnancy or disapproved of personal choices such as natural delivery and breastfeeding [34,35]. There were insufficient alternative arrangements for pregnant surgeons who continued to work through physical discomforts which proved difficult and distressing [32,34,35]. Female surgeons further reported insufficient maternity leave [28,34], compelling them to sacrifice personal vacation or to accept uncompensated leave in exchange

**Table 1. Characteristics of included papers.**

| Author | Year | Country | Participants | | | Methodology | Perspective | |
|---|---|---|---|---|---|---|---|---|
| | | | Number | Gender (Female %) | Age (Mean/ Range) | | Surgical Position (%) | Specialties |
| Hinze *et al.* | 2004 | USA | n = 12 | 58.3 | 31 | Mixed Method; Telephone Surveys, Face-to-face Interviews | Residents (100%) | Internal Medicine, Surgery, Obstetrics and Gynaecology, Anaesthesiology, Dermatology, Ophthalmology |
| Ozbilgin *et al.* | 2011 | UK England | n = 20 | 55.0 | 33–63 | Qualitative; Semi-Structured Interviews | Surgeons (100%) | Internal Medicine, Clinical Pathology, Immunology, Radiology, Surgery, Accident and Emergency Medicine, Endocrinology, Obstetrics and Gynaecology, ENT, Respiratory Medicine |
| Brown *et al.* | 2013 | UK England | n = 17 | 52.9 | 38, 32–48 | Qualitative; Semi-Structured Interviews | Surgeons (100%) | NA |
| Hill *et al.* | 2015 | UK England | n = 10 | 100 | NA | Qualitative; Semi-Structured Interviews | Residents (60.0%), Surgeons (40.0%) | NA |
| Rich *et al.* | 2016 | UK England | n = 137 | 54.0 | NA | Qualitative; Focus Groups; Semi-Structured Interviews | Residents (70.1%), Surgeons (29.9%) | Medicine, Surgery, Psychiatry, General Practice, Clinical Radiology, and Obstetrics and Gynaecology |
| Webster *et al.* | 2016 | Canada | n = 8 | 100 | NA | Qualitative; Focus Groups; Interviews | Surgeons (100%) | NA |
| Dahlke *et al.* | 2018 | USA | n = 98 | 35.7 | 26–30 | Mixed Method; Surveys, Semi-Structured Interviews | Residents (57.1%), Surgeons (42.9%) | General Surgery |
| Myers *et al.* | 2018 | USA | n = 42 | 42.8 | 26–30 | Mixed Method; Surveys; Semi-Structured Interviews | Residents (100%) | General Surgery |
| Yi *et al.* | 2018 | Rwanda | n = 12 | 50.0 | NA | Qualitative; Semi-Structured Interviews | Surgeons (100%) | NA |
| Barnes *et al.* | 2019 | USA | n = 15 | 100 | NA | Mixed Method; Online Surveys, Focus Groups | Residents (100%) | General Surgery, Urology, Neurosurgery, Obstetrics and Gynaecology, Orthopaedic Surgery |
| Liang *et al.* | 2019 | Australia/ New Zealand | n = 12 | 100 | NA | Qualitative; Interviews | Residents (100%) | NA |
| Bernardi *et al.* | 2019 | USA | n = 36 | 38.9 | NA | Qualitative; Survey; Semi-Structured Interviews; Scenario Responses | Residents (27.8%), Surgeons (72.2%) | NA |
| Lu *et al.* | 2019 | USA | n = 63 | 60.8 | 42, 36–52 | Qualitative; Semi-Structured Interviews | Surgeons (100%) | General Surgery, Surgical Oncology, Acute Care Surgery, Cardiothoracic Surgery, Breast Surgery, Vascular Surgery, Colorectal Surgery, Otolaryngology, Plastic Surgery, and Urologic Surgery. |
| Hutchison *et al.* | 2020 | Australia/ New Zealand | n = 46 | 100 | NA | Qualitative; Semi-Structured Interviews | Residents (17.4%), Surgeons (82.6%) | NA |

NA–Not Applicable.

for a longer break [28,32]. Both female and male surgeons perceived maternity leave to be stigmatized due to work-oriented culture with some suggesting that the burden of extra workload resulted in animosity among colleagues [28,30,36].

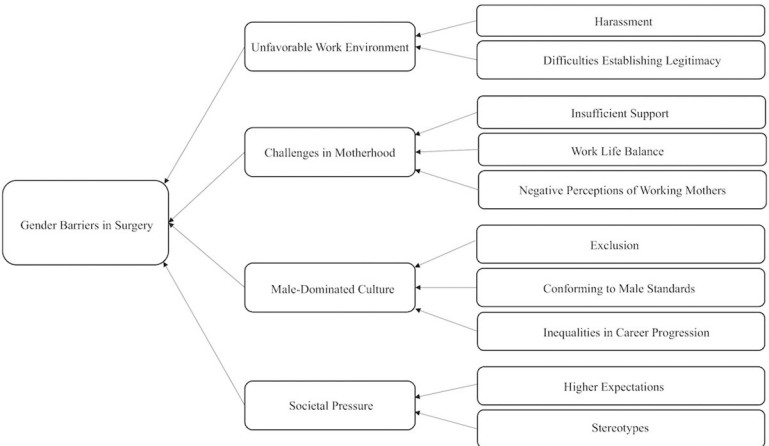

**Fig 2. Mind map of gender barriers faced by female surgical residents and surgeons.**

Female surgeons also observed inequalities in advancements and pay disparity. Women believed that they were being promoted at a slower pace or passed over in favour of male colleagues for referrals despite being equally qualified [28,29,36]. Disparity in pay was either conveniently attributed to women supposedly working less hours due to family commitments or a systemic bias in a historically male-dominated field [28].

**Negative perceptions of females.** Female surgeons and trainees recounted their experiences of being perceived to be less competent or inferior to male counterparts by hospital staff and colleagues alike [25,29,35]. Stereotypes asserted that female surgeons were weaker than their male counterparts, who were thus more suited for the demanding surgical workload [28,37]. Surgeons who were mothers reported strong disapproval and criticism from senior surgeons [25,30,34,35]. Some surgeons stated that being a mother was regarded as a weakness in surgical capabilities, unfairly affecting opportunities, remuneration, reviews, and the ability to apply for leave [28,30,34]. Others shared that mothers who had more familial duties were also perceived to be less committed to work by their seniors, especially if they worked part time, and henceforth, regarded as less respectable [28,31]. Many women also acknowledged that the conflict between family responsibilities and surgical duties resulted in sacrifices of one or the other at times [25,29,30,32–34,37], contributing to feelings of guilt [29,31].

Female surgeons and surgical trainees felt that they were judged by their appearance [27,34,35], which drew attention away from their capabilities and qualities [27]. This resulted in pressure to be extra conscious and intentional about their dressing and upkeep [27,35].

**Lower levels of respect.** Female surgeons and surgical trainees found it harder to command authority as patients and hospital staff tended to dismiss them and trusted more in male colleagues [25,26,35]. Some patients addressed female surgeons and residents inappropriately, with many assuming female surgeons to be nurses [25,26]. Women also shared accounts of being subjected to differential treatment when compared to male peers and receiving less respect from hospital staff [25,26,29].

In addition, some female surgical residents reported that attendings and colleagues addressed them by first names, in contrast to male counterparts who were addressed by titles, which undermined their legitimacy [26]. They recalled sexist remarks from bosses, colleagues, and patients, belittling the skills of female residents, implying their inferiority or negatively assuming women to be averse to challenges [27,30,34,35,37]. They felt that they had to work harder to establish their legitimacy as doctors [25,27,29]. The lack of respect made some female surgeons lose their passion for the specialty [26,30].

## Male-dominated culture

**Exclusion.** Some female and male surgeons reported that an exclusive 'boys' club' existed in surgery [25,28,35]. This was suggested to be due to a historically male-dominated culture resulting in lingering prejudice and subconscious expectations that physicians are male or that females are more suited for less 'masculine' subspecialties [25,26]. Female surgeons felt excluded because they missed out on professional opportunities that transpired in male spaces while others could not participate in a social camaraderie formed solely among male counterparts [26,27,34,35].

**Conforming to male standards.** Female surgeons and surgical trainees often reported having to accept the status quo and adapting to fit into the male culture [24–27,31]. Some felt pressured to tolerate or even engage with lewd remarks by male colleagues to fit in [24,26,34]. Others consciously chose conversational topics that were acceptable and common among male peers [27,31,32]. Female identity was compromised as female surgeons felt that they had to act more masculine in a male-determined surgical field [25,31,37]. Furthermore, male surgeons also shared their difficulties in empathising and understanding challenges faced by women [25], with some who denied the existence of a sex selection criterion in surgery and suggested that women could succeed, solely with the correct mentality [37].

## Societal pressure

**Higher expectations.** Female trainees faced greater pressure as they struggled more to meet higher expectations from hospital staff and seniors, with some elaborating that they had to perform better to be considered equal to their male peers [25,27]. Some women also felt that female seniors who had overcome gender-based challenges in surgery were especially hard on their juniors, expecting them to be able to do the same [34,35]. Female surgeons and residents reported being scrutinized more closely for mistakes [26,29].

**Stereotypes.** Female surgeons perceived innate differences in gender characteristics to contribute to differing expectations and thus, different gender roles [36]. Some also perceived themselves to be less confident than male peers, which held them back in accepting promotions or accepting more leadership roles [25,26,28,37]. This lack of confidence was suggested to be a by-product of working in a male privileged culture [25,26,28]. These stereotypes were further perpetuated by hospital staff, resulting in differential treatment. More menial tasks were often automatically allocated to women over men [25,35], and women observed that they had to behave differently from male peers to achieve the same outcomes [35,36]. Men had a bigger margin to act more unpleasantly, such as raising their voices in the operating theatre or in a more assertive manner, whereas women would be criticized for the same behavior [26,34,36]. Consequently, women had to put in more effort to navigate relationships [35,36].

**Work life balance.** There was gender-based disparity in personal expectations regarding the amount of family responsibilities to take on [25,28,34,37]. Some stated that male surgeons invested more time in work and less time in family [25,28,37], and a larger share of family responsibilities often fell on female surgeons [25,28,34]. They also felt that societal expectations of mothers to take charge of familial duties were imposed on female surgeons [25,36]. Thus, in addition to negative perceptions of motherhood as mentioned previously, personal and societal expectations created additional pressures for female surgeons to juggle family commitments, which negatively impacted their professional lives [28,34].

## Progress towards gender equality

**Gender as a non-issue.** Some female surgeons expressed that gender did not impact their careers [28,36]. Instead, they believed that gender-based difficulties were sometimes results of

individual choices [36]. Others perceived that male surgeons struggled with the same expectations as women [32,36].

**Improvement in gendered culture.** Female surgeons observed drastic improvements in gender equity over the years, as seen by an increasing number of female surgeons and mentors, increased organisational support in terms of ease of leave application, as well as change in perceptions whereby motherhood is more common and thus, less stigmatized, alongside the increasing role of male surgeons in sharing family duties [28,32,36]. Some female surgeons also expressed an unwillingness to be victimized and empowered themselves through a strong belief in their own capabilities [24,37]. They chose to be unaffected by gender issues in both attitude and response [27,35]. Gender equity across specialties was regarded as an ongoing process and one that required time [28,32,36].

**Unique professional traits.** Female surgeons were perceived as detail-oriented, empathetic, and more nurturing towards patients, bringing important skills to surgery that differed from male peers [25,31,35]. They also perceived themselves as less intimidating and more approachable for both patients and peers [31,35]. Female surgeons thought themselves to be preferred and actively chosen by some patients for reasons including more delicate surgical work or being better equipped to understand and look after pregnant patients [35].

## Discussion

This qualitative systematic review sheds light on the lived reality of female surgeons who continue to be subjected to gender discrimination in the forms of unfavorable work environment, male-dominated surgical culture, and societal pressures [38]. In line with the Sex and Gender in Research (SAGER Guidelines), this paper explores gender barriers in context of socially constructed and enacted roles and behaviours which occur in a historical and cultural context, rather than sex-based barriers defined by biological differences between females and males [39]. Despite comparable gender representation in medical school [40], there is a skewed underrepresentation of female surgeons, and negative perceptions of gender bias have been found to significantly reduce interests of female medical students in choosing surgical careers [41,42]. Surgery has also been revealed to be the most women-unfriendly specialty, with the highest number of female physicians changing their area of practice [43]. Hence, there is a need to bring the accounts of these individuals to the forefront and explore programmes to address these inequalities. One such example is the Women in Surgery nationwide program in United Kingdom [44], spearheading the gender equality movement with its extensive resources and a 5000-strong network to connect with surgeons at all levels of training.

Despite improved female representation and support [28,36], females surgeons are still underrepresented in leadership positions, making up only 6.3% of surgical department heads in the United States in 2018 [45]. This translates to a decreased influence in decision making, delaying essential structural reforms that address females' needs and champion gender equality [46,47]. More concerningly, findings suggest survivorship bias, where some senior female surgeons who overcame the odds to succeed in a male-dominated culture expected their subordinates to do the same [34,35]. In such cases, they tended to focus on personal resolve as the crucial success factor, diverting attention from problematic gender barriers. Moreover, exclusion from opportunities in a male-dominated culture [26,27,34,35], was found to extend to operating theatres where females were given less operative autonomy by attendings, impacting their confidence, training quality, and performance [48]. This reinforces existing negative perceptions about their competence [25,29,35], justifying unfair treatment and ultimately presents a self-fulfilling prophecy [49]. To further compound the problem, gender blindness in male surgeons reduced their understanding of barriers that female surgeons face [25], resulting in

inaction to address such issues. The magnitude of gender discrimination may therefore be underestimated and thus, remain deeply entrenched in surgery.

In this review, the views of both trainees and consultants were included, both of which had striking similarities. Females in both groups felt disrespected and subjected to differential treatment by staff, patients and colleagues [25–27,30,34,35,37], suggesting that hierarchy is disregarded and overpowered by stereotypes. Work-life conflict was another common denominator [25,29,30,32–34,37], stemming from deeply-rooted expectations for females to be primary caregivers [25,36], in accordance with existing literature [50,51]. These similarities conclusively indicate a stigmatized culture which fails to improve with seniority, since bullying of younger physicians due to hierarchy and deference [52,53] can be conflated with gender discrimination and accepted as 'rites of passages' during residency [54,55]. This normalises microaggressions [56,57], possibly reinforcing prejudice to create downstream implications for female surgeons. However, manifestations of differences in seniority were observed in harassment incidents, more commonly reported by trainees [24–26,34,35]. This is likely due to a larger power gap, where male attendings, consultants, and patients abused their authority over more vulnerable female trainees, echoed by research with perspectives from victims [58]. Additionally, the incidence of harassment, which is exceptionally high in surgery [59], may in reality be underreported given numerous barriers such as fear of judgment [24], inability to identify sexist behavior [26,27], pressure to fit in [24,26,34], and perceived futility of reporting, which surfaced in similar investigations [60]. Silence and neutrality in such instances may have resulted in repeated occasions of sexual harassment [61], threatening safety in the workplace.

Additionally, the included studies in this paper originated mainly from Western countries [24–36], with only one study conducted in Rwanda which was similar to Western accounts [37], and without literature from the Asian perspective. It is vital to recognise the paucity of qualitative literature in Asia since Asian countries have ideals, gendered culture, and societal norms that are more conservative compared to their more liberal Western counterparts [62]. This is especially apparent in a male-dominated industry such as surgery, where women are more hesitant to voice their opinions about inequalities in surgical training [63]. As a result, it is likely that harassment and gender-based discrimination goes underreported in Asian countries [62], whereas women in Western countries are more vocal about such incidents. Due to the lack of literature, the Asian perspective is underrepresented, where female surgeons may face varying forms or degrees of bias. Hence, findings from this study may not be applicable on a worldwide scale. Further qualitative research needs to be conducted to understand discrimination that Asian female surgeons and trainees face in their line of work. The recognition of gender bias may better help to close the gender gap in surgery and decrease drop-out rates globally.

Even though gender bias is pervasive, there are some who have benefitted from progression towards gender equality [28,32,36], successfully paving their way in the surgical sphere. These female surgeons stay in the field due to their passion [31,37], supportive work environments [32,37], and improvements in gendered culture [29,31,32,37], which proved critical to retention [64,65]. Quality mentorship was especially important [29,31,37], since role models not only allowed female surgeons to visualise future career trajectories, but also convinced them that motherhood can be reconciled with surgery [66,67]. These likely contributed to job satisfaction among female surgeons who stated that they did not regret their choice of specialty [68], or that surgery is a good career for females [69]. Data from the Surgical Infection Society (SIS) also shows that the proportion of female general surgery residents and surgeons has increased from 18.0% in 2000–2005 to 34.6% in 2016–2017, and 15.0% in 2000–2005 to 24.0% respectively, although women in leadership remains greatly underrepresented. [70] Thus,

despite these positive developments, the overwhelming negative evidence reminds readers that gender bias stands to be a deeply concerning phenomenon in surgery.

With the move towards equality in healthcare, several nations have initiated programmes in efforts to reduce gender-based discrimination (Table 2).

These programmes stem from developed nations with literature documenting sexism in surgery, dialling up urgency to address these issues [71–75]. A majority of these programmes aim to tackle male-dominated culture [26,27,34,35], by increasing female representation [71–73,75] and encouraging equal opportunities through female empowerment [72–75]. Another key focus is addressing unfavorable work environments [25,28,34], by providing flexible training options [71–73], resources in reducing harassment [71,74,75], and quality mentorship [71–75]. Supportive work environments, especially in terms of role models, facilitate surgical retainment [32,37,66,67], creating measurable outcomes to justify these programmes. In stark contrast, there were comparatively fewer concrete measures in effecting mindset change, likely because sustained education for all stakeholders is more resource-intensive, time consuming, and its effectiveness difficult to quantify. On this front, the Association of Women Surgeons (AWS) deserves recognition for its commendable efforts in establishing the #HeforShe Committee amidst many other committees, to engage with stakeholders across all stages of training and develop educational toolkits for best gender equity practices [76]. However, apart from AWS which is an international non-profit organization, there remains a lack of robust education measures to promote gender inclusion for national-level programs. The education programme in Australia and New Zealand which only targets surgical fellows [71], fails to dispel ingrained stereotypes across other key stakeholders who contribute to differential treatment and disrespect towards female surgeons [25,26,29,34–36]. Stigma surrounding motherhood [28,30,36], and unequal expectations [25,27], are also two areas that remain largely unaddressed across most programmes.

Although there are pockets of resistance towards changing deeply ingrained surgical culture which may need time to evolve alongside wider societal influences [71], lessons can be learned from other gender equality interventions such as the Athena Scientific Women's Academic Network (SWAN) Charter [77]. Started in 2005 to address gender inequality across science, technology, engineering, mathematics, and medicine (STEMM), this assessment tool grades existing gender bias interventions and gives awards to guide organizations towards practices and policies that advance gender equality [78]. This not only increases visibility and accountability for gender bias issues, but also serves as a signal for cultural shift, providing incentive for innovative solutions, as validated by their 2019 evaluation [79]. It is also important to recognise that more can be done to increase bias literacy across all stakeholders [80] and craft policies that help reduce stigma around maternity leave, while normalising participation of male surgeons in fatherhood duties [81]. This mediates work-life conflict which is the leading cause of attrition for female surgeons [12]. In the age of technology, organizations can also consider leveraging more on the power of social media to create platforms to connect female surgeons with one another, similar to efforts by AWS [82]. Successful outreach has been observed with global movements such as the online campaign #ILookLikeASurgeon which celebrates women in surgery by dismantling the stereotypical image of a surgeon and creating recognition that the appearance, motivations and behaviours of surgeons are as varied as humanity. This movement had a ripple effect, seen in Caprice Greenberg's subsequent reports on gender discrimination, as well as calls from the public for more gender inclusive texts in research [83]. While media advocacy comes with its limitations, it is a viable tool that can spark important dialogues about gender equity within the surgical sphere and beyond. Beyond that, male surgeons also need to be involved as agents of change in the fight for gender equality. In 2019, Dr. Francis Collins, Director of the National Institutes of Health, committed to decline participation in

**Table 2. Overview of programs for gender equality across countries.**

| Country | Name of Program | Organisation | Aims | Objectives and Overview |
|---|---|---|---|---|
| United States of America | Women in Surgery Committee | American College of Surgeons (ACS) | To enable women surgeons of all ages and specialities to develop their individual potential as professionals; promote an environment that fosters inclusion, respect, and success; develop, encourage and advance women surgeons as leaders; and provide a forum and networking opportunities to enhance women's surgical career satisfaction. | • Actively support ACS efforts in Discrimination, Bullying and Sexual Harassment (DBSH)<br>• Provide resources to support women in surgery to take up leadership internationally<br>• Organise the annual Mentorship Programme<br>• Coordinate award nominations and appointments of female leadership in ACS<br>• Identify and submit proposals for presentation at the annual Clinical Congress.<br>• Conduct regular evaluations on committee composition to ensure broad representation and update their mission and goals accordingly. |
| Australia/New Zealand | Building Respect, Improving Patient Safety | Royal Australasian College of Surgeons (RACS) | To build respect in surgery in Australia and New Zealand, and dealing with bullying, discrimination, harassment and sexual harassment. | • Establish a multi-year program to improve complaints management and establish training in DBSH during Fellowship and Surgical Education and Training<br>• Conduct advanced DBSH training<br>• Develop and publish a Diversity Plan, including gender equity, to set expectations for all college activity<br>• Revise accreditation standards for surgical training, ensuring DBSH standards and complaints-resolution are implemented<br>• Incorporate principles recommended by the Expert Advisory Group<br>• Collaborate with various stakeholders to implement recommendations.<br>• Enhance external input and scrutiny of the relevant policies and outcomes. |
| Australia /New Zealand | Women in Surgery Section (WiSS) | Royal Australasian College of Surgeons (RACS) | 1. Encourage and support all Fellow Trainees, particularly females<br>2. Be a source of advice and guidance for Council in relation to gender and trainee issues<br>3. Develop guidelines and policies to combat numerous issues faced by all individuals in the surgical field<br>4. Development of a mentoring program within the College to assist medical students, Trainees and young surgeons | • Support RACS in addressing DBSH and assisted in the formulation of related key policies.<br>• Increase influence of WiSS in RACS committees and address unconscious bias in selection and training<br>• Increase proportion of women applicants, promote women surgeons as positive role models and increase mentorship opportunities.<br>• Actively advocate for availability of flexible training, to enable better work-life balance<br>• Organise various scientific and education events |
| United Kingdom, England | Women in Surgery (WinS) | Royal College of Surgeons of England (RCS) | National initiative dedicated to encouraging, enabling and inspiring women to fulfil their surgical career ambitions. | • Raise awareness of the issues faced by Women Surgeons and devise programmes to support them<br>• Share information on the current situation of women in surgery<br>• Provide sources of support and guidelines on flexible working hours<br>• Organise national events, such as Women in Surgery Conference, to provide support for female surgeons<br>• Encourage more female leadership in the RCS and in surgery via the Lady Estelle Wolfson Emerging Leaders Fellowship |
| Ireland | Progress: Promoting Gender Equality in Surgery | Royal College of Surgeons of Ireland (RCSI) | Promote gender diversity in surgery | • Provide career advice, training opportunities, mentorship and networking resources to encourage students to enter surgery<br>• Increase transparency of fellowship and consultant appointments, increase mentorship and promote better support for personal lives, to build a more inclusive surgical culture for female trainees<br>• Implement policies and programmes to help those with family, balance personal and professional lives<br>• Establish an inclusive environment in professional development, for all surgeons via advocacy of gender equality and providing more support and resources for female surgeons<br>• Publish an annual report to monitor progress on gender diversity initiatives |

*International programs and those related to academic surgery were not included in this table.

high-level conferences comprising of all-male panels [84]. Allyship by those like Dr. Collins sets a precedent on how men can be intentional in advocacy for female representation, sending a strong signal to reform a male-dominated culture [85]. Thus, the importance of including men in diversity efforts cannot be underplayed. Looking forward, there is much space for concrete action to be taken to tackle gender bias in surgery more comprehensively and should be complemented by continued research to cover other gaps in knowledge.

While there are reports suggesting that gender culture in certain specialties, such as Obstetrics and Gynaecology and Ophthalmology [86,87], are increasingly female-dominated, there remains a scarcity of research on gender barriers in these female-dominated fields. This is pertinent as surgery comprises specialties with distinct gender cultures, with literature demonstrating that in Obstetrics and Gynaecology, female surgeons continue to receive unequal compensation while male surgeons experience more patient bias as shown in a meta-analysis [88,89]. Furthermore, some male obstetric surgeons considered their gender to be a limitation and were more likely to change surgical specialty given a choice [90], suggesting that discrimination against male surgeons may be more pervasive in certain specialties. This may go unreported as research illustrates less consistency in labelling discriminatory actions against male victims [91]. The lack of literature may be because female-dominated surgical specialties are a rather recent development [92], and thus have not been subjected to deeper investigations. Consequently, due to the lack of data, this paper highlighted gender bias which can be generalized across most predominantly male surgical specialties. Going forward, further research into gender bias in female-dominated specialties is greatly warranted.

## Limitations

Limitations should be taken into consideration when interpreting these results. Firstly, only articles written in or translated into the English language were included. This systematic review mainly included studies conducted in Western countries with the exception of a single study from Rwanda and may not represent varying cultural contexts. Some studies lacked detail about surgical positions of trainees and consultants to maintain anonymity which could have affected the comparative analysis of experiences between the two groups. Furthermore, due to a lack of literature, this review is unable to explore the impact of discrimination on gender diverse populations and is limited to analysing gender in a binary fashion rather than as a gender spectrum.

## Conclusion

This systematic review sheds light on the numerous gender barriers that continue to stand in the way of female surgeons despite progress towards gender equality over the years. While policy makers have pushed out more measures to address gender bias in surgery, it is important to acknowledge the existing gaps and develop more comprehensive interventions to shape a safe and fair working environment for women, especially working mothers. As the global agenda towards equality progresses, this review serves as a call-to-action to increase the collective effort towards gender inclusivity in surgery, striving towards a field that embraces diversity which will benefit patients in the long run.

## Supporting information

**S1 Checklist.**
(DOCX)

**S1 File. Medline search.**
(PDF)

**S2 File. Qualitative checklist.**
(PDF)

**S3 File. Standards for Reporting Quality Research (SRQR).**
(PDF)

## Acknowledgments

All authors have made substantial contributions to all of the following: (1) the conception and design of the study, or acquisition of data, or analysis and interpretation of data, (2) drafting the article or revising it critically for important intellectual content, (3) final approval of the version to be submitted. No writing assistance was obtained in the preparation of the manuscript. The manuscript, including related data, figures and tables has not been previously published and that the manuscript is not under consideration elsewhere.

## Author Contributions

**Conceptualization:** Choon Seng Chong.

**Data curation:** Wen Hui Lim, Chloe Wong, Sneha Rajiv Jain.

**Formal analysis:** Wen Hui Lim, Chloe Wong, Sneha Rajiv Jain, Cheng Han Ng.

**Investigation:** Chia Hui Tai.

**Methodology:** Wen Hui Lim, Chloe Wong, Cheng Han Ng.

**Project administration:** Wen Hui Lim.

**Supervision:** Cheng Han Ng, M. Kamala Devi, Dujeepa D. Samarasekera, Shridhar Ganpathi Iyer, Choon Seng Chong.

**Validation:** Chia Hui Tai.

**Writing – original draft:** Wen Hui Lim, Chloe Wong, Sneha Rajiv Jain, Cheng Han Ng.

**Writing – review & editing:** Wen Hui Lim, Chloe Wong, Sneha Rajiv Jain, Cheng Han Ng, Chia Hui Tai, M. Kamala Devi, Dujeepa D. Samarasekera, Shridhar Ganpathi Iyer, Choon Seng Chong.

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
