## [Decision Letter · Decision Letter 0]

22 Dec 2020

PONE-D-20-37354

The unspoken reality of gender bias in surgery: A qualitative systematic review

PLOS ONE

Dear Dr. Chong,

Thank you for submitting your manuscript to PLOS ONE. After careful consideration, we feel that it has merit but does not fully meet PLOS ONE’s publication criteria as it currently stands. Therefore, we invite you to submit a revised version of the manuscript that addresses the points raised during the review process.

We look forward to receiving your revised manuscript.

Kind regards,

Leonidas G Koniaris, MD

Academic Editor

PLOS ONE

Journal Requirements:

Additional Editor Comments (if provided):

Please address minor comments in the reviews. We llok forward to accepting this excellent manuscript.

Reviewers' comments:

Reviewer's Responses to Questions

**Comments to the Author**

1. Is the manuscript technically sound, and do the data support the conclusions?

Reviewer #1: Yes

Reviewer #2: Yes

2. Has the statistical analysis been performed appropriately and rigorously? 

Reviewer #1: Yes

Reviewer #2: N/A

3. Have the authors made all data underlying the findings in their manuscript fully available?

Reviewer #1: Yes

Reviewer #2: Yes

4. Is the manuscript presented in an intelligible fashion and written in standard English?

Reviewer #1: Yes

Reviewer #2: Yes

5. Review Comments to the Author

Reviewer #1: In this manuscript, the authors perform a systematic review of gender bias and harassment in the discipline of surgery. I applaud the authors for this study, which is a much needed one in order to unify the message of the need for a change in the male-dominated culture of surgery. The study was well executed, the manuscript was well written and has a clear message.

MINOR

1. There are several large-size studies on gender bias (Ann Thorac Surg 2020; 109(1):14-17. n=663), sexual harassment (Ann Thorac Surg 2020; 109(4):1283-1288. n=790), and salary disparities (Ann Thorac Surg 2020; S0003-4975(20)31687-8. n=1069) in cardiothoracic surgery. Including these references would help to amplify your message.

2. On the topic of motherhood and work-life balance, did the authors note the manuscript of gender differences in academic surgery and work-life balance of 127 faculty and 116 trainees (J Surg Res 2017; 218:99-107). Inclusion of this reference also would contribute significantly to the message of this manuscript.

3. Line 74 (“gender disparities that cripple the progression of female surgeons.” Use of the word “cripple” is overly dramatic and the word renders a negative reaction. Making overstatements, such as this one, may turn away the audience and, hence, the message of this very important manuscript is lost. Recommend using “stunts” or “impedes” or a more neutral term that conveys the same thought.

4. Line 224. “Camaraderie” is spelled incorrectly.

Reviewer #2: Overall assessment

This is an important topic and the paper has done a good job of synthesizing the literature and discussing its relevance. Some improvements to terminology and clarity will enhance the work.

Strengths

1. important topic

2. generally easy-to-read with a few minor typographical issues

3. thorough discussion

Areas for Improvement

Major issues

1. confusion of sex and gender throughout the text

2. lack of clarity and whose voice is being represented when talking through the results

Minor and editorial issues

please do not start sentences with numbers unless you spell them out.

The methods are sparsely described in the abstract into should include key issues like how many people screened/extracted data and how synthesis was completed.

In your introduction you state that gender discrimination has thwarted interest from women candidates and that is why women are underrepresented - what about bias in selection of candidates you know if the success rates are similar in terms of admission.

The paper talks about 2 genders men and women but ignores gender diverse populations and how they have been treated in surgery. It may be this is due to a lack of literature or a lack of intention of addressing this issue- please clarify.

It is also not clear whether you are talking about sex or gender I would’ve assumed gender… But you talk about males and females which is sex so please use appropriate language for sex or gender.

Be clear about whether you are stating a fact or the perceptions of the respondents in the qualitative studies. For example lines 179/180 Women were promoted at a slower pace or passed over in favour of male colleagues for referrals despite being equally qualified [25, 26, 33]. You have not made it clear whether this is the perception of individuals or based on data. I realize I could go look up each of these references and try to figure out what they say but I think you have the responsibility of being clear whether you are conveying perceptions from the qualitative literature which represent the thoughts and experiences of individuals or whether you are looking at quantitative data that reflects an analysis of what is happening. They may or may not be the same.

Similarly it is often not clear whose voice you are representing from the qualitative literature. Example: others who had more familial duties were also perceived to be less committed to work, especially if they worked part time, and henceforth, regarded as less respectable… Lines 192/193… It is not clear whether it is qualitative analysis of the perceptions of supervising physicians and staff about the women surgeon,,, or it is the women surgeon’s perceptions of how they think the physicians and staff receive them… Again potentially related but potentially not and it is very important in qualitative research to be clear about whose views are being represented. I assume that you have included studies that have different perspectives and the perspective must be clear both in your chart of the included studies and when you are representing themes. Another example - female surgeons were also preferred and actively chosen by some patients for reasons including more delicate surgical work or being better equipped to understand and look after pregnant patients [32]…. It is not clear if you are analysing the perspectives of patients or how surgeons think patients have interacted with them

When you are talking about gender blindness he specifically state sex differences but I think your paper is about gender differences since it is societal roles and perceptions not biology that is determining the themes that you are addressing. Again, I think the language in this paper around sex and gender is mixed up- please consult Sager guidelines

Start your discussion with an overall summary of the key contribution of the work in terms of findings not a statement of claim … this is the first qualitative systematic review ….

It is certainly true that there is an underrepresentation of women but there is as you point out considerable efforts to change this and I think it would be good not only to say that underrepresentation exists but to have some data in your paper about how quickly this is changing. In other words are we on track for even representation in their near future or is the rate of change so slow that this underrepresentation is still not resolving.

6. PLOS authors have the option to publish the peer review history of their article (what does this mean?). If published, this will include your full peer review and any attached files.

Reviewer #1: No

Reviewer #2: No

---

## [Author Response · Author response to Decision Letter 0]

18 Jan 2021

Reviewer 1: Dear Reviewer, thank you for the comments. We have amended the manuscript to include the mentioned references. 

Reviewer 2: Dear Reviewer, thank you for the comment. We have clarified the respective terms and added in a clearer definition of gender which is the main scope of our paper. We have also amended the results section to better reflect the statements of the respondents. We have edited the limitations in covering gender diverse populations in this study. All other minor editorial issues have also been addressed accordingly.

---

## [Editor Report · Decision Letter 1]

20 Jan 2021

The unspoken reality of gender bias in surgery: A qualitative systematic review

PONE-D-20-37354R1

Dear Dr. Chong,

We’re pleased to inform you that your manuscript has been judged scientifically suitable for publication and will be formally accepted for publication once it meets all outstanding technical requirements.

Kind regards,

Leonidas G Koniaris, MD

Academic Editor

PLOS ONE

Additional Editor Comments (optional):

There are a number of typos that should be corrected.
---

## [Editor Report · Acceptance letter]

22 Jan 2021

PONE-D-20-37354R1 

The unspoken reality of gender bias in surgery: A qualitative systematic review 

Dear Dr. Chong:

I'm pleased to inform you that your manuscript has been deemed suitable for publication in PLOS ONE. Congratulations! Your manuscript is now with our production department. 

Kind regards, 

on behalf of

Dr. Leonidas G Koniaris 

Academic Editor

PLOS ONE